# Air Stagnations for China (1985–2014): Climatological Mean Features and Trends

Qianqian Huang, Xuhui Cai, Yu Song, Tong Zhu

College of Environmental Sciences and Engineering, State Key Joint Laboratory of Environmental Simulation and Pollution Control, Peking University, Beijing, 100871, China

*Correspondence to:* Xuhui Cai (xhcai@pku.edu.cn)

**Abstract.** Air stagnation is an important meteorological measure for unfavourable air pollution conditions, but little is known about it in China. We conducted a comprehensive investigation of air stagnation in China from January 1985 to December 2014, based on sounding and surface observations of 81 stations. The stagnation criteria were revised to account for the large topographical diversity in this country. It is found that the annual mean air stagnation occurrences are closely related to general topography and climate features. Two basins in the northwest and southwest of China—Tarim and Sichuan Basins—exhibit the most frequent stagnation occurrence (50% days per year), whereas two plateaus (Tibet-Qinghai and Inner Mongolia Plateau) and the east coastal areas experience the least (20% days per year). Over the whole country, air stagnations achieve maxima in summer and minima in winter, except for Urumqi, a major city in the northwest of China, where stagnations keep a rather constant value yearly around with a minimum in spring. There is a nationwide positive trend in stagnation occurrence during 1985–2014, with the strongest increasing centres over Shandong Peninsula in eastern China and the south of Shaanxi in central China. Changes in air stagnation occurrences are dependent on three components (upper- and lower-air winds, precipitation-free days). It shows that the behaviours of upper-air wind speeds are main drivers of the spatial distribution and trend of air stagnations, near-surface winds the next, and dry days contribute the least.

## 1 Introduction

Air quality is strongly dependent on meteorological state, which controls the transport and dispersion of air pollutants within the lower atmosphere. The meteorological state with lingering anticyclones and persistent calm winds leading to poor ventilation and no precipitation to wash out pollutants is defined as an air stagnation event (Wang and Angell, 1999). It has been observed that stagnation events are usually related to air pollution episodes

(Jacob et al., 1993; Wang and Angell, 1999; Mickley et al., 2004; Wu et al., 2008; Fiore et al., 2012). For example, ozone measurements from rural sites in the eastern and western United States in 1978 and 1979 indicated that the majority of ozone episodes occurred during air stagnant conditions (Logan, 1989). The stagnation of air masses led to an enhancement of surface ozone and CO mixing ratios over Western/Central Europe (Ordónez et al., 2010;

Leibensperger et al., 2008), and a 2.6 $\mu g$ m$^{-3}$ increase of fine particulate matter in the United States (Tai et al., 2010). The sensitivity of air quality to stagnations has been investigated by perturbing meteorological variables in regional chemical transport models (Liao et al., 2006). Jacob and Winner (2009) collected and compared results from different perturbation studies about the effects of weather conditions on ozone and particulate matter concentrations, and summarized that air stagnation demonstrates a robust positive correlation. However, actual air

pollution level is affected by not only meteorological conditions, but also other complex factors such as emission sources and chemical reactions (Cao et al., 2007; Guo et al., 2009; He et al., 2001; Yang et al, 2016). With other factors unknown, the air stagnation index may show poor correlation to actual air pollution data, in certain situations. The strength of air stagnation index is that, it provide an independent view to the meteorological background relevant to air pollution, without being interfered by the complexity of other factors such as the

variation of source emissions (Horton et al., 2014).

Air stagnation is identified by thresholds of daily upper- and lower-air winds and precipitation (Wang and Angell, 1999). The "upper-air winds" here refer to winds at about 5 km above the ground. From a meteorological perspective, this level is important because of its connection to near-surface synoptic systems. It is found that the

20 movement of surface cyclones tend to travel in the direction of the upper flow at roughly a quarter to half of the speed (Frederick et al., 2012). These kinds of near-surface synoptic systems are essential to air pollution (Jacob and Winner, 2009; Cai et al., 2017). On the other hand, near-surface winds and precipitation determine dilution and washout of air pollutants, both of them are also relevant to practical air quality. Therefore, the air stagnation is a relatively simple, but conceptually robust metric to air pollution.

In previous works by Korshover (1967) and Korshover and Angell (1982) for the United States, air stagnation events were defined using daily weather maps from the US National Weather Service. They are periods with (i) the surface geostrophic wind is less than 8 m s$^{-1}$, generally corresponding to a 10m wind speed less than 3.2 m s$^{-1}$ (Wang and Angell, 1999); (ii) the wind speed at 500 hPa is less than 13 m s$^{-1}$; (iii) no precipitation. Wang and

30 Angell (1999) followed this metric but replaced the dataset with US National Centres for Environment Prediction

(NCEP)/National Centre for Atmospheric Research (NCAR) reanalysis archive (2.5 °× 2.5 °). With this dataset, they studied the climatology of air stagnation for the United States from 1948 to 1998 and found that air stagnation events happen most frequently in the southern states during an extended summer season from May to October. Based on the work of Wang and Angell (1999), the National Climatic Data Center (NCDC) monitors air stagnation days in the United States with a finer gridded reanalysis data (0.25 °× 0.25 °) and provides maps about air stagnation distribution every month (http://www.ncdc.noaa.gov/societal-impacts/air-stagnation/). Following the NCDC's metric, Leung and Gustafson (2005) examined the potential effects of climate change on U.S. air quality by analyzing the simulated changes in stagnation events during 2045–2055. Horton et al. (2012, 2014) furthered this work and used a multi-model ensemble to project the future air stagnation occurrence on a global scale. They found that global warming could be expected to result in increasing stagnation frequency over the eastern United States, Mediterranean Europe and eastern China.

China has experienced rapid economic growth and industrialization in the recent decades and become the second largest energy consumer in the world (Chan and Yao, 2008). Tremendous energy consumption results in heavy air pollution. Research on its air quality is indispensable and studies of the relevant meteorological states are essential. Up to now, to the authors' knowledge, studies about air stagnation generally focused on the United States or on a global scale. Our primary purpose in the current study is to investigate the climatological mean features and trends of air stagnations for China, based on 30-year (1985–2014) observations on stations across the country.

**2 Data and Method**

**2.1 Data**

Long-term (1985–2014) dataset of daily-mean surface wind speeds (observed at 10 m above the surface) and daily precipitation data were obtained from China Meteorological Administration (CMA). These data are available at website (http://data.cma.cn/data/detail/dataCode/SURF_CLI_CHN_MUL_DAY_CES_V3.0.html). Upper-air wind speeds were obtained from Wyoming University soundings database (http://weather.uwyo.edu/upperair/sounding.html). This database provides twice daily (0000 and 1200 UTC) atmospheric soundings from stations that participate in global data exchange. Daily averages of upper-air wind speed at mandatory levels of 500, 400, and 300 hPa were used here.

We obtained datasets of all the radiosonde stations across China (95 stations) and two stations (Blagovescensk and Vladivostok) outside but near the border of the country. Among them, 66 stations have corresponding surface datasets from CMA (See Appendix A). For each of the other 31 radiosonde stations, we considered the average of surface stations within 150 km as a substitute. In this way, we got additional 15 stations (See Appendix B). Air stagnations of these 81 stations are analyzed in this study. Figure 1 displays the distribution of the stations. It shows that there are relatively less stations in Tibet-Qinghai Plateau, particularly in the western Tibet. Besides of this, the 81 stations cover all the contiguous China well.

### 2.1.1 Quality Control

Surface datasets have been quality controlled by CMA (http://data.cma.cn/data/detail/dataCode/SURF_CLI_CHN_MUL_DAY_CES_V3.0.html). Upper-air winds data were eliminated from this analysis if

$$\left| U_i - \overline{U} \right| > 2\sigma ,\tag{1}$$

where $U_i$ denotes the $i$ th upper-air wind speed at a given mandatory pressure level of a certain station, $i$ ranges from 3 to $n$-2 and $n$ is the total number of the data sample. $\overline{U}$ and $\sigma$ are calculated as follows:

$$\overline{U} = \frac{1}{5} \sum_{i=i-2}^{i+2} U_i ,\tag{2}$$

$$\sigma = \sqrt{\frac{\sum_{i=i-2}^{i+2} \left( U_i - \overline{U} \right)^2}{4}} ,\tag{3}$$

Subjective quality control procedure was also applied. Time series plot of upper-air wind speeds at each station was drawn to screen temporally inhomogeneous data. Under these two quality control procedures, no upper-air wind data have been considered abnormal and removed. Therefore, we consider the data used in this study are reliable.

### 2.1.2 Data Completeness

A general survey shows that datasets of 73 stations are available from January 1985 to December 2014, while the other 8 stations (Wenjiang, Jinghe, Chongqing, Shanghai, Vladivostok, Yuzhong, Zhangqiu, Qingyuan) cover less than 30 years. The shortest duration is 7 years and 3 months (200710–201412) at Jinghe station. The percentage of valid data (data of upper- and lower-air wind and precipitation all valid) of each station is

summarized in Appendices A and B. It is shown that for all stations except Blagovescensk, more than 95% of the data are valid. Overall, the datasets are sufficient to conduct a climatological research about air stagnations over China.

## 2.2 Method

We adjust the NCDC air stagnation index and a given day is considered stagnant when daily total precipitation is < 1 mm (i.e. a dry day), daily-mean surface wind speed is < 3.2 m s$^{-1}$ and upper-air wind speed is < 13 m s$^{-1}$. In previous studies, the upper-air wind is defined as the wind at 500 hPa. For China, however, this criterion is not appropriate for its great physical diversity. Particularly, the Tibet-Qinghai Plateau has the average height of over 4000 m. The wind speeds at 500 hPa are not a good representative to the upper-air winds above ground. Therefore, we refined the criteria to be topographically dependent and the mandatory level to provide upper-air wind is chosen according to the station's elevation (Table 1).

With the modified criteria, air stagnation days are identified by checking meteorological conditions of every day at each station. Furthermore, if there are 4 or more consecutive days of air stagnation conditions at a given station, those days are considered as one air stagnation case (Wang and Angell, 1999). Results of stagnation days and cases were interpolated with cubic splines to 2 °× 2 °grid to show the spatial distribution over continental China.

## 3 Results

### 3.1 Annual Occurrence

Annual mean air stagnation days are distributed with substantial regional heterogeneity (Fig. 2a). They are most prevalent over basins in the northwest and southwest of China (i.e., Xinjiang and Sichuan provinces) where air stagnant conditions account for 50% of days per year on average, and less prevalent (about 33% days per year) over the northernmost and southernmost of the country. The remainder domains of China experience even less stagnant days, especially Tibet-Qinghai Plateau, Inner Mongolia Plateau and the east coastal areas, where stagnant conditions are the least (less than 20% days per year). The distribution of stagnation cases agrees well with that of stagnation days (Fig. 2b). The strongest stagnant centers—Xinjiang and Sichuan basins—exhibit more than 16 cases per year, while the weakest centers—Tibet-Qinghai and Inner Mongolia Plateau and east coastal areas—only experience less than 2 cases per year. Air stagnation cases usually persist about 5 days in a

majority of these areas (Fig. 2c). The ones over basins of Xinjiang and the southern China last longer (6 days). The longest duration of air stagnation conditions occurs in the south of Guangxi, lasting more than 7 days.

## 3.2 Seasonal Occurrence

Generally, most air stagnant conditions happen during summer season while only a few of those occur during winter. Stagnation days in autumn are slightly more than those in spring (Fig. 3). Similar feature was also found in the earlier work of Wang and Angell (1999) for the United States. The seasonal variation of stagnation is attributed to seasonal shift of pressure patterns and general circulation. A much weaker pressure gradient in summer is a well-known seasonal feature in mid-latitudes (Frederick et al., 2012). This feature is very evident in upper layer atmosphere in China (Ding et al., 2013, their Fig. 1.1). However at sea-level surface, the case in eastern Asia and China are complicated by the sub-tropical high in the east and the continental low and in the west respectively. As a result, Asia summer monsoon prevails in eastern China. Though for this, the sea level pressure gradient in summer is still much weaker than that in winter (Ding et al., 2013, their Fig. 1.1). A weaker wind in both upper and surface layer accompanies the weaker pressure gradient, and results in more air stagnant occurrence in China and North America (Wang and Angell 1999).

We choose four stations (Harbin, Urumqi, Beijing and Chongqing shown in Fig. 1 as triangles) as well as the average of entire China (all stations in this study) to demonstrate the seasonal variation of air stagnation days and cases. Figure 4 shows that for all but Urumqi, stagnation days and cases begin to increase in March or May, then grow dramatically and achieve maxima in July or August, then fall sharply and reach minima in December or January. However, monthly stagnation days and cases of Urumqi show much small variation in a year with their minima in April. This may attributes to the unique local climate there. Xinjiang basin is isolated at the center of the continent, far away from the coast, and blocked by the huge Tibet-Qinghai Plateau in the south. Therefore, the East Asia monsoon, particularly the East Asia summer monsoon, which is prevailing in east of China, influences little to Xinjiang. As a result, the seasonal feature of the stagnation in Urumqi is different from that in east of China. By comparing between these stations, we find that stagnations over Chongqing, Beijing and Urumqi are more than the average level of the entire country. Among them, Chongqing station has the largest variation of stagnations, and Beijing the next. Urumqi keeps a relatively high stagnation frequencies throughout the year.

### 3.3 Trend of Stagnations

The majority of China exhibits positive trends about 10–20 days and 1–3 cases per decade in stagnation days and cases, respectively (Fig. 5a, b). The strongest center located in Shandong Province and the south of Shaanxi, with rising rates of more than 20 days and 3 cases per decade. Only four fragments exhibit weak decrease: the extreme north of China, regions located in the south of Gansu and north of Sichuan, the westernmost and the southernmost part of China. The negative trend of stagnation varies from 0 to 10 days and 0 to 1 case per decade over the first three regions, and 30 days and 5 cases per decade over the last region. We have assessed the statistical significance of above results and 52% of stations have passed the 0.05 significance test. Not only stagnation frequencies show an increase over large areas, the duration of stagnation cases also exhibits a nationwide extension about 0.3 day decade$^{-1}$ (Fig. 5c). Only a few scattered regions show a gradually shortened stagnation duration, including the extreme north of China, Yangtze River Delta, and the westernmost and southernmost region.

Stagnation trends of four stations (Harbin, Urumqi, Beijing and Chongqing) are also discussed specifically, as well as the average results of the entire country. Figure 6 shows that all four stations exhibit positive trends in air stagnation days, ranging from 3 to 14 days decade$^{-1}$, and the national-averaged increasing rate is about 6 days decade$^{-1}$. Moreover, the two stations experiencing more stagnant days (Urumqi and Chongqing) exhibit a slower rising rate about 5 and 3 days decade$^{-1}$, respectively, whereas the other two stations having relatively less stagnations (Harbin and Beijing) show a faster rate about 14 and 8 days decade$^{-1}$, respectively.

### 4 Discussions

Air stagnation is identified based on the thresholds of three components: the lower- and upper-air wind speeds and precipitation-free days. Analysis of each individual component is helpful in understanding the behaviour of stagnations. Figure 7a shows that the distribution of annual mean upper-air wind field is reversely correlated to that of stagnation occurrences. The upper-air wind is relatively weak in the northeast of China, Tarim Basin, Sichuan Basin and the southernmost of the country, which are exactly the same four regions exhibiting frequent stagnations. Different from the pattern of upper-air wind field, surface wind field exhibits a strong wind center (> 3 m s$^{-1}$) around the northeast of China, and a weak one (about 1.5 m s$^{-1}$) in Sichuan Basin (Fig.7b). Surface wind speed over the remainder of China is around 2 m s$^{-1}$. The distribution of dry days (daily total precipitation

< 1 mm) is largely related to the latitude. Figure 7c shows that generally dry days in the north is more than that in the south. But, specifically, the south of Xinjiang Province experiences maximum dry days of more than 350 days per year, while the south of Shaanxi Province shows the minimum about 200 days per year.

We further analyse the degrees of stagnation's dependence on every individual component (Fig. 8). It implies that 76% ($R^2$) of the spatial variation in stagnant days can be explained by the distribution of upper-air wind fields. The correlation even reaches as high as 82% in autumn (Fig. S1 in the Supplement). The surface wind field only accounts for 20% and the spatial distribution of dry days barely influences the stagnation variation. Figure 7 and 8 show that stagnation occurrences result from the cumulative responses of individual stagnation components, but

the distribution of upper-air wind speed exerts the dominant influence. The same feature was also suggested in the global research by Horton et al. (2012) in the area of China.

Similarly, we examine the relationship of stagnation trends with each component (Fig. 9) and find that the pattern of trends in upper-air wind field is similar to that of stagnant conditions. Decreases in upper-air wind field

substantially outnumber increases throughout the country (Fig. 9a), and the regions showing rapidly decreasing winds coincide with those exhibiting robust growing stagnations in Fig. 7. Trends in surface wind field and dry days may show a slightly different pattern from trends in stagnation (Fig. 9b, c), but they still contribute more or less for some regions. For areas with increasing stagnations like the northeast and the south of Shaanxi province, dry days show a substantial positive trend (about 3–7 days decade$^{-1}$) and upper- and lower-air wind speeds show

a remarkable negative trend (about 0.3–0.6 m s$^{-1}$ decade$^{-1}$). For Shandong region, both upper- and lower-air wind speeds exhibit a substantial decrease of more than 0.3 m s$^{-1}$ decade$^{-1}$, although the dry days show a slightly decrease about 1 day decade$^{-1}$. To summarize, the stagnation trends are contemporaneous effects of two or three components.

The dependence of national averaged stagnation trends on every individual component are shown in Fig. 10. It can be seen that the negative trend in upper-air wind speed accounts for 73% of the increase in stagnant days. The ratio varies slightly with seasons and the highest one (79%) occurs in spring (Fig. S2 in the Supplement). Inter-annual variations in surface wind speed and dry days explain 42% and 32%, respectively. Still, trend in upper-air wind is the dominant contributor.

The trends of these three components (upper-air winds, near-surface winds and daily precipitation) are driven by climate change. The decrease in upper-air winds is resulted from smaller contrasts of the sea level pressure and a weakened Hadley circulation, both as a consequence of global warming (Lau et al., 2006; Lu et al., 2007; Seidel et al., 2008). Near-surface wind decline is attributed to the slowdown in atmospheric general circulation (Guo et al., 2011; Xu et al., 2006; Vautard et al., 2010) and stabilized atmosphere by light absorbing aerosols (Li et al., 2016; Peng et al., 2016; Wang et al., 2013). The decreasing number of rainy days, due to suppressed light rainfall but intensified heavy rainfall, is mainly attributed to the accumulation of greenhouse gases and aerosols in China (Gong et al., 2004; Liu et al., 2015; Wang et al., 2011a; Wang et al., 2016). To sum up, climate changes alter atmospheric circulation and the hydrological cycle, which influence the occurrences of air stagnations—the meteorological background of air quality.

Stagnation is a meteorological metric for potential air pollution occurrence. Once there exist anthropogenic or natural air pollutants, they are liable to accumulate and result in poor air quality over regions that experience frequent air stagnant conditions. In contrast, over regions with infrequent stagnations, air pollutants will quickly be transported far away and diluted. Current results of the prevalent centers of stagnation days and cases are consistent with areas of heavy pollutions in China (Mamtimin and Meixner, 2007; Wang et al., 2011b; Chen and Xie, 2012; Liu et al., 2013; Zhang et al., 2014; Li et al., 2015). The spatial distribution of annual mean visibility during 1985–2014 (Fig. S3 in the Supplement) shows that regions of Sichuan basin, the west of Xinjiang and North China Plain are the centres exhibit low visibility. This feature corresponds well to the frequent air stagnation occurrences in Fig. 2. The correlation between a time series of air stagnation days and visibility over the whole country is −0.69 during 1985–2014 (Fig. S4 in the Supplement). It means that the air stagnation does correlate negatively to visibility, in general. It confirms that stagnation is an effective metric to measure the potential unfavourable meteorological states on air quality.

It should be noted that air stagnation metric does not take into account emissions or atmospheric chemical reactions. So there may be discrepancies between variations of stagnation index and actual air pollutant concentrations in certain situations. For example, aerosol concentration in China is characterized by high value in winter and lower one in summer. This observational fact is obviously related to the seasonal variation of source emission, since there is more coal consumption in winter for heating, particularly in north China (Cao et al., 2007; He et al., 2001). To make this kind of meteorological metric applicable for practical air pollution forecasting,

Yang et al. (2016) incorporate source emission information into their PLAM index, and improve the forecasting skill successively. In this work, we aim to analyse the general features of meteorological background relevant to air pollution by means of the air stagnation metric, without concerning more about the complexity of source emissions or chemical reactions.

## 5 Conclusions

Based on upper and surface wind speeds and daily precipitation data from 81 stations across the country, this paper presented climatological mean values and trends of air stagnations for China, in the period from January 1985 to December 2014. The dependence of stagnations on three components (upper- and lower-air winds and dry days) was examined. A topographically dependent version of air stagnation criteria was applied to account for the terrain effect in China.

Annual mean of air stagnation occurrence varies spatially, in agreement with topography and climate features. Two basins in the northwest and southwest of China—Tarim and Sichuan Basins—exhibit the most frequent stagnation occurrence (50% days per year), while two plateaus (Tibet-Qinghai and Inner Mongolia Plateaus) and the east coastal areas experience the least (20% days per year). Seasonal variation of air stagnations is also presented. For a general view of the whole country, stagnations happen the most frequently in summer and the least in winter. For specific stations of Harbin, Beijing and Chongqing, stagnations vary with months dramatically and achieve maxima in July or August and minima in December or January, whereas stagnations in Urumqi keep a rather constant value with minimum in April.

There is a nationwide positive trend in stagnation days and cases, as well as case duration, in 1985–2014. The strongest increasing centres are located over Shandong Province in the eastern China and the south of Shaanxi in the central of the country. Only two exceptional regions—the southernmost and westernmost part of China—exhibit a negative trend in both occurrence and duration.

Stagnation occurrence contemporaneously response to three components, i.e., upper- and lower-air winds and precipitation-free days. Among them, upper-air wind speed plays a dominant role, explaining 76% and 73% of the spatial distribution and trend of air stagnations respectively. The lower-air wind exerts a minor influence.

These results are corroborated by the global research from Horton et al. (2012). The spatial variation of dry days barely influences that of stagnations, whereas their inter-annual variability explains 32% of stagnation trend.

Air stagnation climatology presents a specific view to the natural background of atmosphere features being responsible to air pollution levels. The results presented in this paper may have significant implication to air pollution research, and may be used in atmospheric environment management or air pollution control.

**Acknowledgements**

This work is partially supported by National Natural Science Foundation of China (41421064, 41575007), Clean Air Research Project in China (Grant Nos. 201509001, 201409001), and National Key Technology Research and Development Program (2014BAC06B02). The authors would like to acknowledge CMA and Wyoming University for providing the long-term and public available surface observations and sounding data. The first author would like to thank Mr. Jian Wang for helping pre-process the sounding data. Two anonymous reviewers' comments and suggestions are greatly acknowledged.

**Competing interests:**

The authors declare that they have no conflict of interest.

**Appendix A**

**The 66 radiosonde stations that have corresponding surface datasets from CMA. The periods for which radiosonde and surface datasets were both available are summarized with the start and end dates in the column date range. Data periods less than 30 years are highlighted in bold. The percentages of valid data are also presented here.**

| Station | ID* | Latitude (°N) | Longitude (°E) | Elevation (m) | Date Range | Valid Data |
|---------|-----|---------------|----------------|---------------|------------|------------|
| Hailar | 50527 | 49.21 | 119.75 | 611 | 198501–201412 | 97.8% |
| Nenjiang | 50557 | 49.16 | 125.23 | 243 | 198501–201412 | 98.9% |
| Harbin | 50953 | 45.75 | 126.76 | 143 | 198501–201412 | 99.4% |
| Altay | 51076 | 47.72 | 88.08 | 737 | 198501–201412 | 99.2% |
| Yining | 51431 | 43.95 | 81.33 | 664 | 198501–201412 | 99.3% |
| Urumqi | 51463 | 43.77 | 87.62 | 919 | 198501–201412 | 99.2% |
| Kuqa | 51644 | 41.71 | 82.95 | 1100 | 198501–201412 | 99.4% |
| Kashi | 51709 | 39.46 | 75.98 | 1291 | 198501–201412 | 99.2% |
| Ruoqiang | 51777 | 39.02 | 88.16 | 889 | 198501–201412 | 99.1% |
| Hotan | 51828 | 37.13 | 79.93 | 1375 | 198501–201412 | 99.4% |
| Hami | 52203 | 42.81 | 93.51 | 739 | 198501–201412 | 99.3% |
| Dunhuang | 52418 | 40.15 | 94.68 | 1140 | 198501–201412 | 99.1% |
| Jiuquan | 52533 | 39.75 | 98.48 | 1478 | 198501–201412 | 98.9% |
| Minqin | 52681 | 38.63 | 103.08 | 1367 | 198501–201412 | 98.5% |
| Golmud | 52818 | 36.4 | 94.9 | 2809 | 198501–201412 | 99.2% |
| Dulan | 52836 | 36.29 | 98.09 | 3192 | 198501–201412 | 96.9% |
| Xining | 52866 | 36.71 | 101.75 | 2296 | 198501–201412 | 99.1% |
| Erenhot | 53068 | 43.65 | 112 | 966 | 198501–201412 | 99.1% |
| Hohhot | 53463 | 40.81 | 111.68 | 1065 | 198501–201412 | 99.3% |
| Yinchuan | 53614 | 38.47 | 106.2 | 1112 | 198501–201412 | 98.2% |
| Taiyuan | 53772 | 37.77 | 112.55 | 779 | 198501–201412 | 99.1% |
| YanAn | 53845 | 36.59 | 109.5 | 959 | 198501–201412 | 99.3% |
| Pingliang | 53915 | 35.54 | 106.66 | 1348 | 198501–201412 | 98.9% |
| XilinHot | 54102 | 43.95 | 116.05 | 991 | 198501–201412 | 99.2% |
| Tongliao | 54135 | 43.6 | 122.26 | 180 | 198501–201412 | 99.3% |
| Changchun | 54161 | 43.9 | 125.21 | 238 | 198501–201412 | 99.0% |

**(Appendix A Continued)**

| Station | ID* | Latitude (°N) | Longitude (°E) | Elevation (m) | Date Range | Valid Data |
|---|---|---|---|---|---|---|
| Chifeng | 54218 | 42.25 | 118.95 | 572 | 198501–201412 | 99.3% |
| Yanji | 54292 | 42.88 | 129.46 | 178 | 198501–201412 | 99.4% |
| Shenyang | 54342 | 41.75 | 123.43 | 43 | 198501–201412 | 99.2% |
| Linjiang | 54374 | 41.71 | 126.91 | 333 | 198501–201412 | 98.9% |
| Beijing | 54511 | 39.93 | 116.28 | 55 | 198501–201412 | 99.4% |
| Dalian | 54662 | 38.9 | 121.62 | 97 | 198501–201412 | 99.3% |
| Lhasa | 55591 | 29.65 | 91.12 | 3650 | 198501–201412 | 97.5% |
| Yushu | 56029 | 33 | 97.01 | 3682 | 198501–201412 | 97.1% |
| Hezuo | 56080 | 35 | 102.9 | 2910 | 198501–201412 | 98.4% |
| Garze | 56146 | 31.61 | 100 | 522 | 198501–201412 | 98.7% |
| Wenjiang | 56187 | 30.7 | 103.83 | 541 | **200407–201412** | 98.1% |
| Xichang | 56571 | 27.9 | 102.26 | 1599 | 198501–201412 | 98.7% |
| Tengchong | 56739 | 25.11 | 98.48 | 1649 | 198501–201412 | 99.1% |
| Kunming | 56778 | 25.01 | 102.68 | 1892 | 198501–201412 | 99.0% |
| Simao | 56964 | 22.76 | 100.98 | 1303 | 198501–201412 | 99.3% |
| Mengzi | 56985 | 23.38 | 103.38 | 1302 | 198501–201412 | 99.2% |
| Zhengzhou | 57083 | 34.7 | 113.65 | 111 | 198501–201412 | 99.3% |
| Hanzhong | 57127 | 33.06 | 107.02 | 509 | 198501–201412 | 99.4% |
| Jinghe | 57131 | 34.26 | 108.58 | 411 | **200710–201412** | 100.0% |
| Enshi | 57447 | 30.28 | 109.46 | 458 | 198501–201412 | 99.4% |
| Yichang | 57461 | 30.7 | 111.3 | 134 | 198501–201412 | 99.3% |
| Wuhan | 57494 | 30.61 | 114.12 | 23 | 198501–201412 | 99.4% |
| Chongqing | 57516 | 29.51 | 106.48 | 260 | **198708–201412** | 98.2% |
| Guiyang | 57816 | 26.47 | 106.65 | 1222 | 198501–201412 | 98.0% |
| Guilin | 57957 | 25.33 | 110.3 | 166 | 198501–201412 | 99.4% |
| Ganzhou | 57993 | 25.85 | 114.94 | 125 | 198501–201412 | 99.1% |
| Xuzhou | 58027 | 34.27 | 117.15 | 42 | 198501–201412 | 99.3% |
| Nanjing | 58238 | 32 | 118.8 | 7 | 198501–201412 | 99.4% |
| Shanghai | 58362 | 31.4 | 121.46 | 4 | **199106–201412** | 98.3% |

**(Appendix A Continued)**

| Station | ID[*] | Latitude (°N) | Longitude (°E) | Elevation (m) | Date Range | Valid Data |
|---|---|---|---|---|---|---|
| Anqing | 58424 | 30.53 | 117.05 | 20 | 198501–201412 | 99.4% |
| Hangzhou | 58457 | 30.22 | 120.16 | 43 | 198501–201412 | 99.3% |
| Nanchang | 58606 | 28.6 | 115.91 | 50 | 198501–201412 | 99.1% |
| QuXian | 58633 | 28.95 | 118.86 | 71 | 198501–201412 | 99.1% |
| Fuzhou | 58847 | 26.07 | 119.27 | 85 | 198501–201412 | 99.3% |
| Xiamen | 59134 | 24.47 | 118.08 | 139 | 198501–201412 | 99.3% |
| Baise | 59211 | 23.9 | 106.6 | 175 | 198501–201412 | 99.2% |
| Wuzhou | 59265 | 23.48 | 111.3 | 120 | 198501–201412 | 98.3% |
| Shantou | 59316 | 23.35 | 116.66 | 3 | 198501–201412 | 99.2% |
| Nanning | 59431 | 22.62 | 108.2 | 126 | 198501–201412 | 99.3% |
| Haikou | 59758 | 20.03 | 110.34 | 24 | 198501–201412 | 99.2% |

* World Meteorological Organization Identification Number.

**Appendix B**

**Same as Appendix A, but for the 15 radiosonde stations and corresponding surface stations within 150 km.**

| Station | ID | Latitude (°N) | Longitude (°E) | Elevation (m) | Surface Stations (CMA) | Latitude (°N) | Longitude (°E) | Distance (km) | Date Range | Valid Data |
|---|---|---|---|---|---|---|---|---|---|---|
| Blagovescensk | 31510 | 50.52 | 127.5 | 177 | 50353 | 51.43 | 126.4 | 127.58 | 198501–201412 | 65.9% |
| | | | | | 50564 | 49.26 | 127.2 | 141.64 | | |
| Vladivostok | 31977 | 43.26 | 132.05 | 82 | 54096 | 44.23 | 131.1 | 132.61 | **199408–201412** | 96.1% |
| | | | | | 59287 | 23.1 | 113.2 | 132.7 | | |
| KingsPark | 45004 | 22.3 | 114.16 | 66 | 59293 | 23.44 | 114.4 | 129.34 | 198501–201412 | 99.1% |
| | | | | | 59501 | 22.48 | 115.2 | 110.8 | | |
| YuZhong | 52983 | 35.87 | 104.15 | 1875 | 52884 | 36.21 | 103.6 | 65.18 | **200107–201412** | 94.5% |
| Linhe | 53513 | 40.75 | 107.4 | 1041 | 53336 | 41.34 | 108.3 | 100.66 | 198501–201412 | 99.0% |
| | | | | | 54725 | 37.3 | 117.3 | 70.05 | | |
| Zhangqiu | 54727 | 36.7 | 117.55 | 123 | 54823 | 36.36 | 117 | 59.92 | **200308–201412** | 98.3% |
| | | | | | 54843 | 36.45 | 119.1 | 142.05 | | |
| Qingdao | 54857 | 36.06 | 120.33 | 77 | 54843 | 36.45 | 119.1 | 117.68 | 198501–201412 | 99.3% |
| Weining | 56691 | 26.86 | 104.28 | 2236 | 57707 | 27.18 | 105.2 | 95.07 | 198501–201412 | 98.4% |
| Nanyang | 57178 | 33.03 | 112.58 | 131 | 57265 | 32.23 | 111.4 | 141.85 | 198501–201412 | 99.2% |
| | | | | | 57290 | 33 | 114 | 133.37 | | |

**(Appendix B continued)**

| Station | ID | Latitude (°N) | Longitude (°E) | Elevation (m) | Surface Stations (CMA) | Latitude (°N) | Longitude (°E) | Distance (km) | Date Range | Valid Data |
|---|---|---|---|---|---|---|---|---|---|---|
| Changsha | 57679 | 28.2 | 113.08 | 46 | 57687 | 28.13 | 112.6 | 52.52 | 198501–201412 | 98.6% |
| Huaihua | 57749 | 27.56 | 110 | 261 | 57745 | 27.27 | 109.4 | 66.58 | 198501–201412 | 99.2% |
| Sheyang | 58150 | 33.75 | 120.25 | 7 | 58040 | 34.5 | 119.1 | 136.93 | 198501–201412 | 99.3% |
|  |  |  |  |  | 58251 | 32.52 | 120.2 | 136.88 |  |  |
| Fuyang | 58203 | 32.86 | 115.73 | 33 | 58102 | 33.52 | 115.5 | 77.52 | 198501–201412 | 99.3% |
|  |  |  |  |  | 58221 | 32.57 | 117.2 | 143.98 |  |  |
| Shaowu | 58725 | 27.32 | 117.45 | 219 | 58715 | 27.35 | 116.4 | 104.76 | 198501–201412 | 99.1% |
|  |  |  |  |  | 58834 | 26.39 | 118.1 | 121.86 |  |  |
| QingYuan | 59280 | 23.66 | 113.05 | 19 | 59082 | 24.41 | 113.4 | 89.16 | **199601–201412** | 95.3% |
|  |  |  |  |  | 59287 | 23.1 | 113.2 | 64.13 |  |  |
|  |  |  |  |  | 59293 | 23.44 | 114.4 | 140.78 |  |  |

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

| Station Elevation (m) | Topographically dependent upper-air wind speed criterion |
|---|---|
| 0–1000 | wind speed at 500 hPa < 13 m s$^{-1}$ |
| 1000–3000 | wind speed at 400 hPa < 13 m s$^{-1}$ |
| 3000–4000 | wind speed at 300 hPa < 13 m s$^{-1}$ |

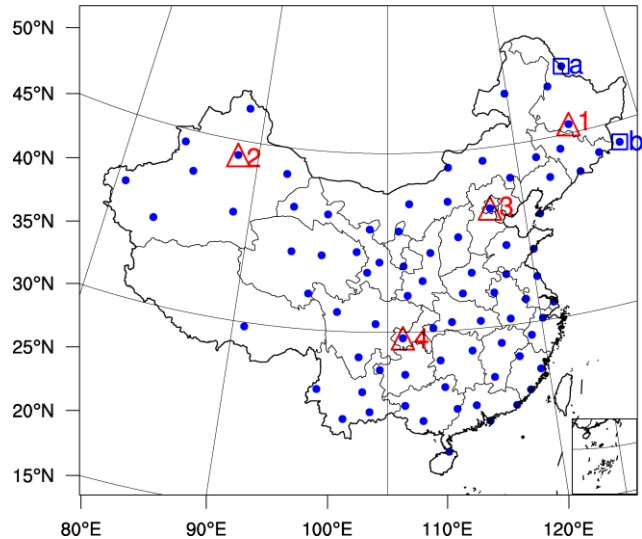

**Figure 1. Distribution of observation stations. Triangles indicate four stations selected to discuss seasonal variations of air stagnations in Section 3: 1 for Harbin, 2 for Urumqi, 3 for Beijing, 4 for Chongqing. Squares indicate two stations outside of China: Blagovescensk (a) and Vladivostok (b).**

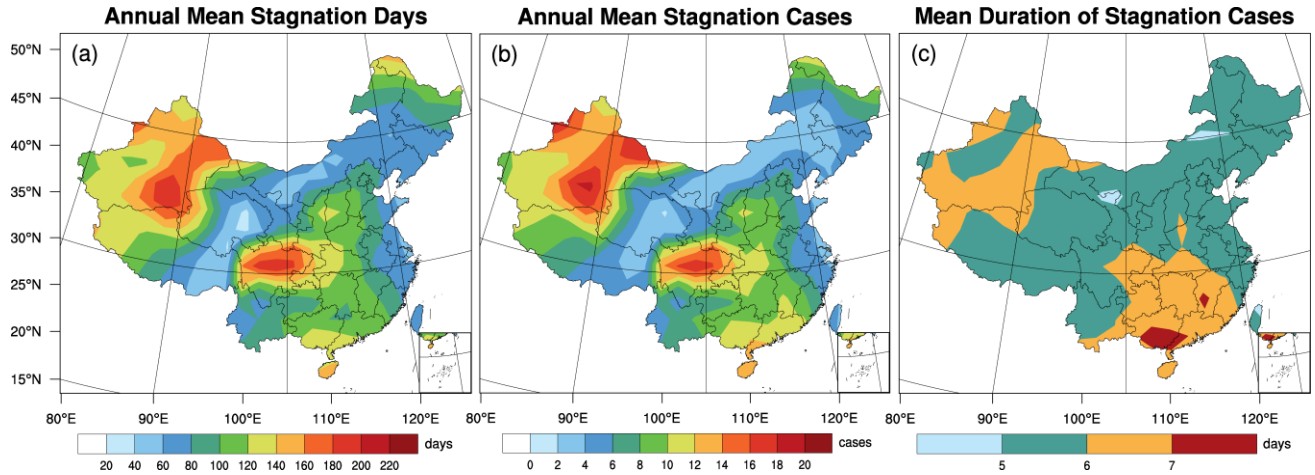

**Figure 2. Annual mean air stagnation days (a) and cases (b) and mean duration of stagnation cases in days (c) for China (1985–2014).**

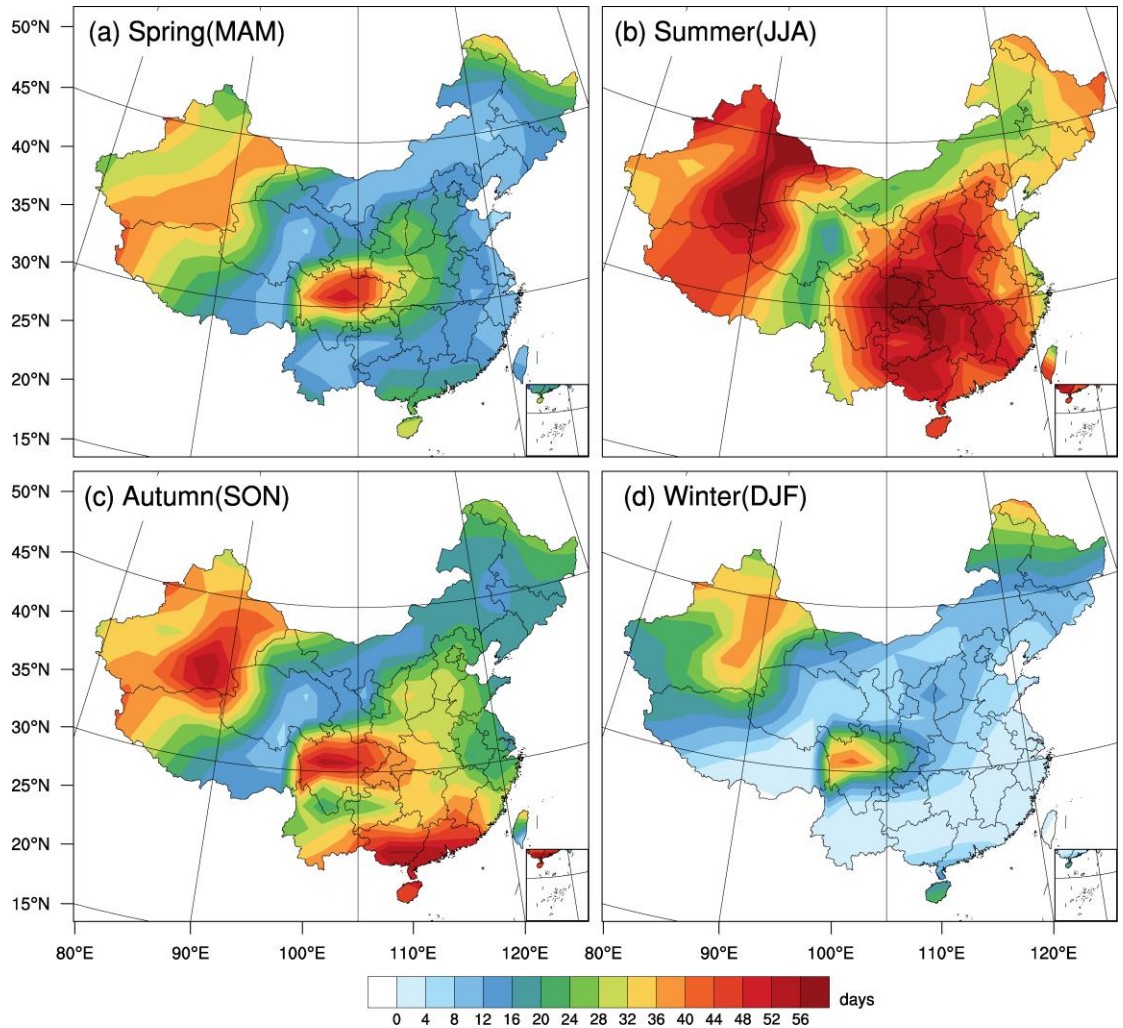

**Figure 3. Average air stagnation days in spring (a), summer (b), autumn (c) and winter (d), 1985–2014.**

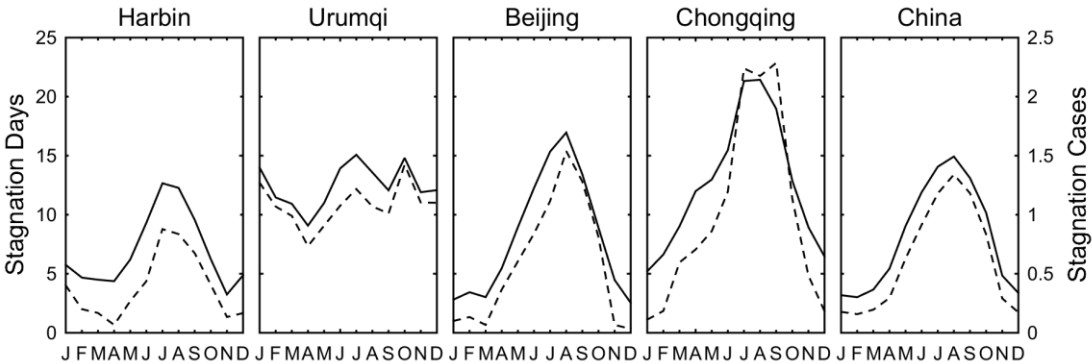

**Figure 4. Seasonal cycles of monthly mean air stagnation days and cases for four stations (Harbin, Urumqi, Beijing and Chongqing) and the entire China. Solid line: stagnation days; dashed line: stagnation cases.**

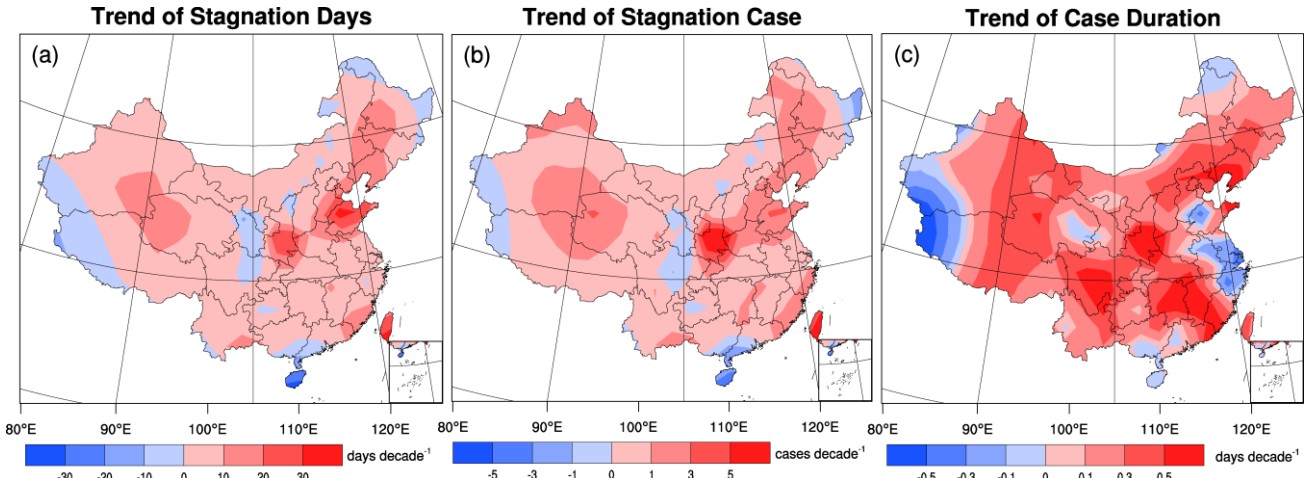

**Figure 5. Trends of stagnation days (a), cases (b) and the duration of stagnation cases (c), 1985–2014.**

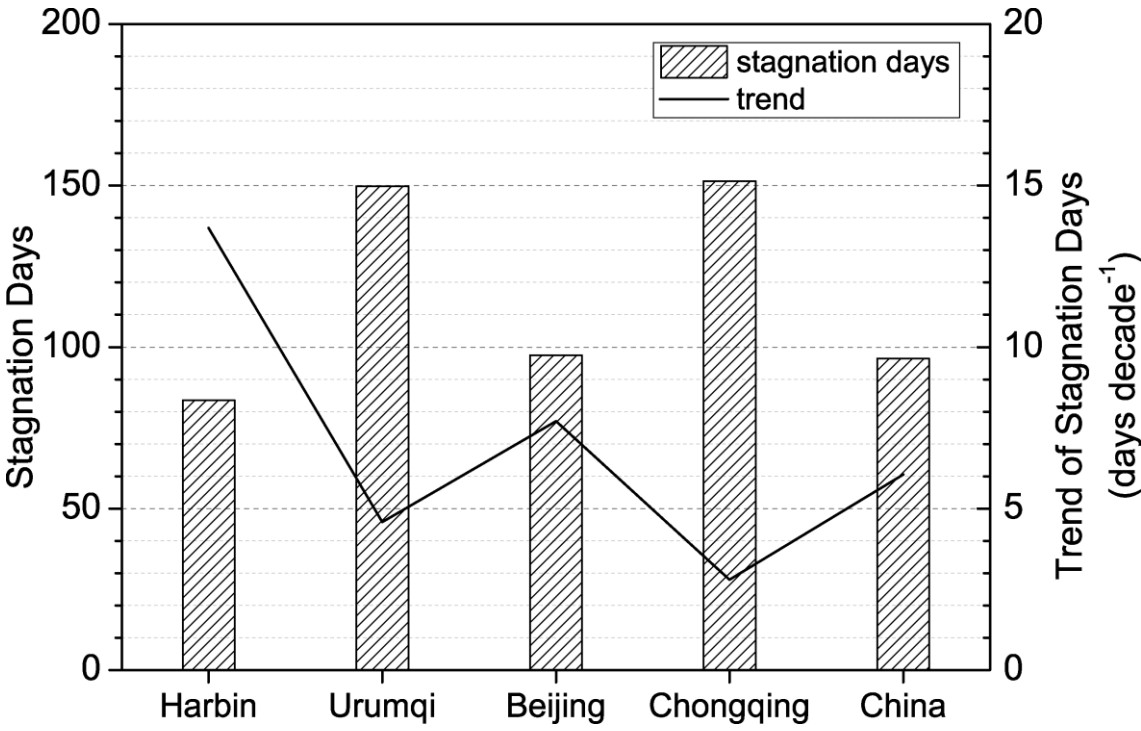

**Figure 6. Annual mean stagnant days and the corresponding trend at four stations (Harbin, Urumqi, Beijing and Chongqing) and for the whole country.**

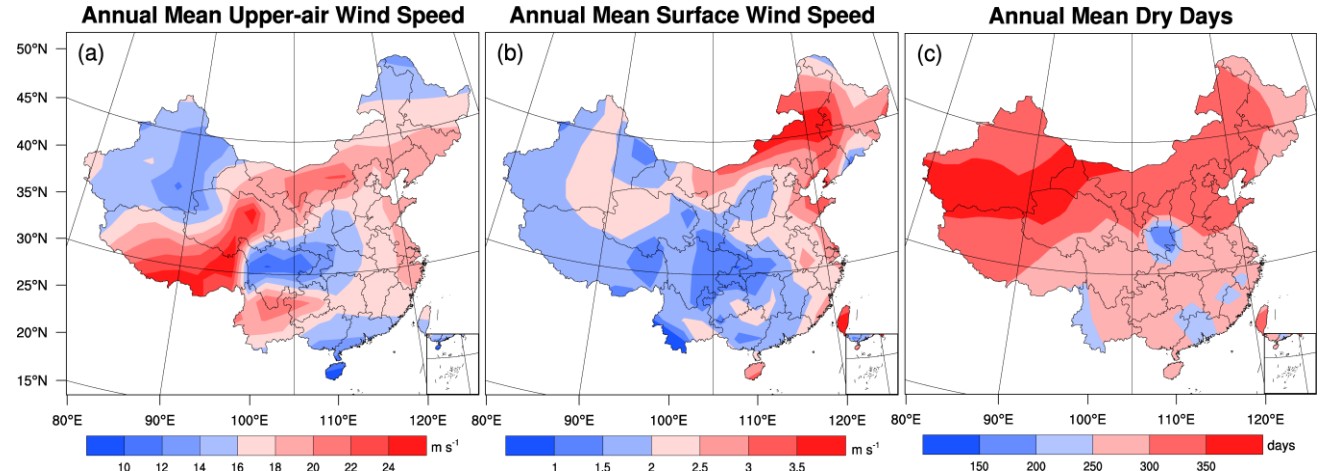

**Figure 7. Annual mean upper-air wind speed (a), surface wind speed (b) and dry days (c) in China (1985–2014).**

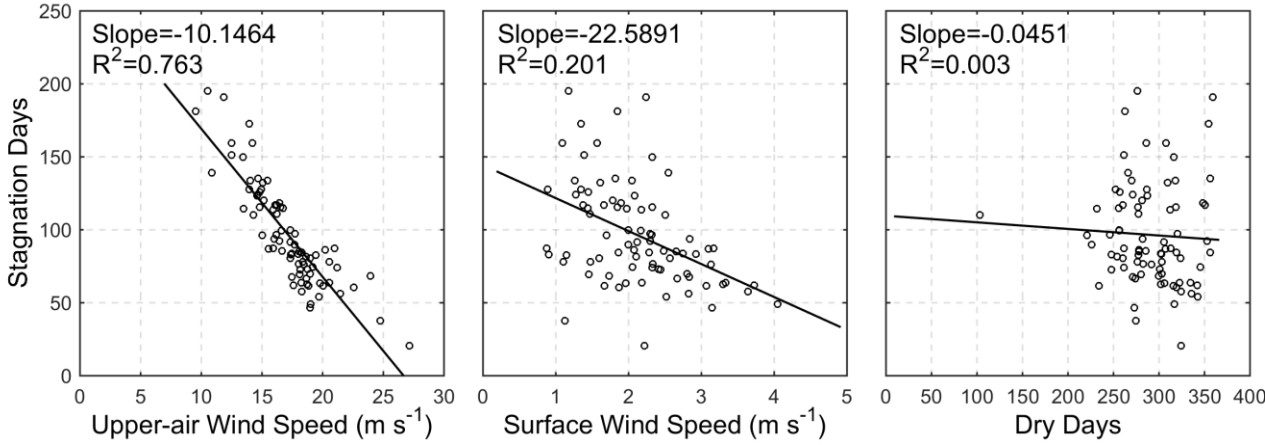

**Figure 8. Dependence of spatial distribution of stagnation days on three components (upper-air wind speed, surface wind speed and dry days). Linear regression coefficients between annual mean stagnation days at 81 stations and each corresponding component are shown.**

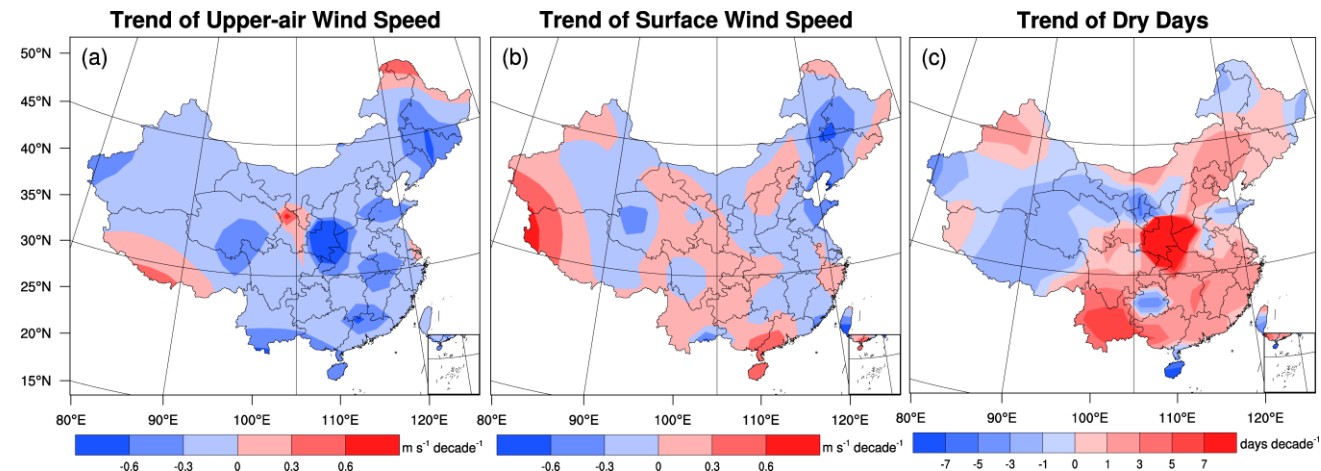

**Figure 9. Trends of upper-air wind speed (a), surface wind speed (b) and dry days (c), 1985–2014.**

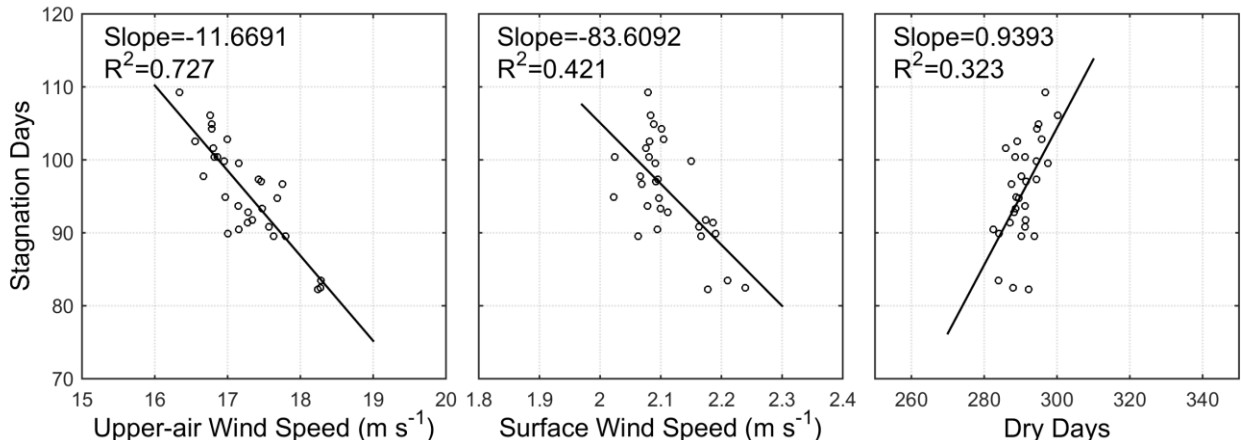

**Figure 10. Same as Fig. 8, but for the trends of stagnant days. Linear regression coefficients between national averaged stagnant days in 30-year period and each corresponding component are shown.**