# Peer review of "Air Stagnations for China (1985–2014): Climatological Mean Features and Trends"

_Atmospheric Chemistry and Physics, 2016_

## Referee Comment (RC1) · Anonymous Referee #1 · 18 Jan 2017

General comments:

This manuscript presents very intriguing results with regard to the current status and positive trend of air stagnation in China during the previous 30 years. The analysis method is sounding and structure is well organized. This work certainly fit the scope of ACP. Overall, the results obtained here gain additional insight into how meteorology modulates the air pollution that is blanket across China. However, traditional, the atmospheric stability is characterized by thermodynamic metrics, such as PLAM developed by Yang et al. ACP 2016, which has been demonstrated to sufficiently reflect the real air pollution condition at most regions of China. The stagnation index (SI) was improved in this study by taking the terrain into account, which is composed of three components, but by nature it is quite different metric for characterize the atmospheric stability highly associated with air pollution. As such, more discussions are required,

as detailed in the following comments. Therefore, I recommend this work be published in ACP after the following concerns have been adequately addressed.

Specific comments:

1. The seasonality of stagnation index (SI) with maximum in summer but minima in winter reported here seems contrary to the actual air pollution. For example, a large amount of previous studies (e.g., Cao et al., JGR 2007; Guo et al., AE 2009; He et al., AE 2011) have shown that seasonal variation of aerosol concentrations is significant in China, characterized by highest concentration in the winter and lowest in the summer. More interestingly, Guo et al. (2009) revealed more complicated seasonality of aerosol concentrations based on 11 PM observations sites. Therefore, the appropriate discussion concerning how the SI connects with aerosol pollution on the monthly scale is needed, given the easy availability nowadays. In other words, I would like to see the scatter plots of Stagnation days versus PM2.5 or PM10, similar to the plot convention in Fig. 8. 2. Page 3, line 13-17: For each of the other radiosonde stations: it is better to clarify how many station, for example 8 other radiosonde stations? According to Guo et al., ACP 2016, there are 120 radiosonde stations across China, and actually only part of the whole radiosonde netword was used in this study. 3. Figure 1: The site outside of China like Blagovescensk, Vladivostok, and KingsPark should be plotted and marked. 4. Page 4, line 25: What kind of spatial interpolation methods have the authors applied? More details should be given, like the size of grid point for the maps. 5. Section 3.2:" During summer a weaker pressure gradient lingers over China, which results in weaker atmospheric circulations and more frequent stagnations." , i think at least one figure is needed to corroborate this argument, e.g., the pressure gradient difference between summer and annual mean.

Rerences: Cao et al., 2007.Spatial and seasonal distributions of carbonaceous aerosols over China. JGR, 112, D22. Guo, J.P., X. Zhang, H. Che, S. Gong, X. An, C.X. Cao, J. Guang, H. Zhang, Y.Q. Wang, X.C. Zhang, P. Zhao, X.W. Li: Correlation between PM Concentrations and Aerosol Optical Depth in Eastern China, Atmospheric

[Figure]

Environment, 43(37): 5876-5886. 2009. Guo, JP, et al. The climatology of planetary boundary layer height in China derived from radiosonde and reanalysis data. Atmos Chem Phys 2016c; 16(20), 13309-13319. doi:10.5194/acp-16-13309-2016. He et al., 2011. The characteristics of PM2.5 in Beijing, China. Atmos. Environ. 35 (29): 4959–4970. Yang, Y.Q., et al. 2016. PLAM – a meteorological pollution index for air quality and its applications in fog-haze forecasts in North China, Atmos. Chem. Phys., 16, 1353-1364,

---

## Referee Comment (RC2) · Anonymous Referee #2 · 22 Feb 2017

This study explores the long-term surface meteorology and sounding measurements in China to study the climatology and inter-decadal trends of air stagnation in China since the 1980s. Considering the current severe haze pollution in China, this topic is interesting to the community and the compiled data sets are valuable for other pollution-related researches. Therefore, the materials in the manuscript are definitely publishable. However, my major concern is that, in contrast to the abundance of data involved in this study, the analyses of the data and the exploration of physics behind the observed relationships are quite inadequate. The authors should carefully address this issue before I can recommend its publication on ACP. Some detailed comments are provided below.

1) Section 3.2, the seasonality of air stagnation is interesting, but the authors' explanation and analysis are insufficient. For example, authors simply attributed the peak

of stagnation in the summer to the "weak pressure gradient", but failed to explain what cause the weak pressure gradient in the summertime Eastern China. How is it linked to the sub-tropical high or the monsoon circulation? Answers can be found by looking at reanalysis of meteorological fields.

2) Similarly, it is interesting to see the larger influence of upper-level wind on air stagnation than that of the surface wind (this point should be highlighted in the abstract), but the physical relationship between upper-level wind and air stagnation is vaguely described in the manuscript. It should be thoroughly discussed in the introduction part. Is the upper-level wind here associated with moving speed of high pressure systems?

3) Inter-decadal trends of air stagnation in China are derived in this study, but what drive these trends are not discussed in the paper. Greenhouse gas, anthropogenic aerosol, or some other forcing agents? I understand that this complex question may be beyond the main scope of the paper, but some discussions along this line could largely sharpen the climatic implication of this study. I suggest a more thorough literature review about recent climate changes in China, in terms of hydrological cycle and precipitation extremes (Wang Y. et al., 2016, JGR), monsoon circulations (Li Z., et al., 2016, Rev. Geos.), etc, and the authors can link the trends to the possible factors at play.

4) Would the increasing trend of air stagnation be partly caused by the accumulation of the absorbing aerosols in boundary layer and a consequent stabilization effect (Wang Y., et al., 2013, AE; Peng J. et al., 2016, PNAS)?

5) The correlation analyses in Fig. 8 and 10 use annual mean values. Why not monthly mean? That would give you more dots and larger statistical significance.

6) What is the correlation between the air stagnation and visibility measurements in China since the 1980s? I assume they are highly correlated, but it is still interesting to see their relationships in time series and spatial distribution in China.

---

## Author Comment (AC1) · 10 Apr 2017

**Response to Referee #1**

Manuscript: Air Stagnations for China (1985–2014): Climatological

Mean Features and Trends (acp-2016-1072)

We are very grateful to the referees for their insightful comments. We carefully consider all the comments and suggestions, and carry out additional analysis on relevant problems. We hope that we have taken most of the referees' concerns into the revised manuscript. All the comments of Referee #1 are copied below in italics, and followed by our responses.

**Referee #1:**

*General comments:*

*This manuscript presents very intriguing results with regard to the current status and positive trend of air stagnation in China during the previous 30 years. The analysis method is sounding and structure is well organized. This work certainly fit the scope of ACP. Overall, the results obtained here gain additional insight into how meteorology modulates the air pollution that is blanket across China. However, traditional, the atmospheric stability is characterized by thermodynamic metrics, such as PLAM developed by Yang et al. ACP 2016, which has been demonstrated to sufficiently reflect the real air pollution condition at most regions of China. The stagnation index (SI) was improved in this study by taking the terrain into account, which is composed of three components, but by nature it is quite different metric for characterize the atmospheric stability highly associated with air pollution. As such, more discussions are required, as detailed in the following comments. Therefore, I recommend this work be published in ACP after the following concerns have been adequately addressed.*

**Response:**

We thank the referee for the positive evaluation to this article. As the referee mentioned, there are different metrics related to air pollution. Among them, the PLAM is of solid theoretic bases and is also successfully applied in China to analyze and forecast air pollution (Yang et al., 2016). The air stagnation index (SI) is a different metric and, to our knowledge, has not yet been systematically applied in China. So, in the revised manuscript, we try to add more discussions on relevant questions. The difference between SI and PLAM is also remarked.

References:

Yang, Y. Q., Wang, J. Z., Gong, S. L., Zhang, X. Y., Wang, H., Wang, Y. Q., Wang, J., Li, D., and Guo, J. P.: PLAM - a meteorological pollution index for air quality and its applications in fog-haze forecasts in North China, Atmos. Chem. Phys., 16, 1353-1364, doi:10.5194/acp-16-1353-2016, 2016.

*Specific comments:*

*1. The seasonality of stagnation index (SI) with maximum in summer but minima in winter reported here seems contrary to the actual air pollution. For example, a large amount of previous studies (e.g., Cao et al., JGR 2007; Guo et al., AE 2009; He et al., AE 2011) have shown that seasonal variation of aerosol concentrations is significant in China, characterized by highest concentration in the winter and lowest in the summer. More interestingly, Guo et al. (2009) revealed more complicated seasonality of aerosol concentrations based on 11 PM observations sites. Therefore, the appropriate discussion concerning how the SI connects with aerosol pollution on the monthly scale is needed, given the easy availability nowadays. In other words, I would like to see the scatter plots of Stagnation days versus PM2.5 or PM10, similar to the plot convention in Fig. 8.*

**Response:**

Air stagnation index (SI) has long been studied in North America continent (as introduced in the manuscript). This is a simple and meaningful metric to air pollution. However, this metric indicates adverse meteorological condition to air pollution, rather than the air pollution itself. Because air pollution level is determined by not only meteorological conditions, but also other complex factors such as emission sources and chemical reactions (Cao et al., 2007; Guo et al., 2009; He et al., 2011; Yang et al, 2016), etc. So, this kind of metric is only an indicator for air pollution potential—the potential for air pollution events. This is the reason that the metric may show bad correlation or even contrary to actual air pollution, when comparing them **directly**.

For the fact that aerosol concentration in China is characterized by high value in winter and lower one in summer, it is obviously related to the seasonal variation of source emission, since there is more coal consumption in winter for heating, particularly in north China (Cao et al., 2007; He et al., 2011). Therefore, to make this kind of metric applicable for practical air pollution forecasting, Yang et al. (2016) incorporate source emission information into their PLAM index, and improve the forecasting skill successively. On the other hand, one can also investigate the sensitivity of air quality to stagnations by perturbing meteorological variables in regional chemical transport models (Liao et al., 2006). Jacob and Winner (2009) collected and compared results from different perturbation studies about the influence of meteorological variables on ozone and particulate matter concentrations, and summarized that air stagnation is the only metric that consistently shows a robust positive correlation.

The purpose of this paper is to provide a general distribution of air pollution potential over China, as well as its long-term trend, by the metric of air stagnation. This is also an effort to "isolate the meteorological factors that produce poor air quality" (Horton et al., 2012). The strength of this work is that, it provide an independent view to the meteorological background relevant to air pollution, without being interfered by the complexity of other factors, e.g. the variation of source emissions.

Since our purpose is not to apply directly to air pollution diagnosis or forecasting, like the work of Yang et al. (2016), and previous works have demonstrated good correlation between air stagnation and air pollution (Jacob and Winner, 2009), therefore at least at this stage, we do not try to compare the metric with concentrations of air pollutants.

This discussion has been added in the revised manuscript.

References:

Cao, J. J., Lee, S. C., Chow, J. C., Watson, J. G., Ho, K. F., Zhang, R. J., Jin, Z. D., Shen, Z. X., Chen, G. C., Kang, Y. M., Zou, S. C., Zhang, L. Z., Qi, S. H., Dai, M. H., Cheng, Y., and Hu, K.: Spatial and seasonal distributions of carbonaceous aerosols over China, J. Geophys. Res.-Atmos., 112, doi:10.1029/2006JD008205, 2007.

Guo, J. P., Zhang, X. Y., Che, H. Z., Gong, S. L., An, X. Q., Cao, C. X., Guang, J., Zhang, H., Wang, Y. Q., Zhang, X. C., Xue, M., and Li, X. W.: Correlation between PM concentrations and aerosol optical depth in eastern China, Atmos. Environ., 43, 5876–5886, doi:10.1016/j.atmosenv.2009.08.026, 2009.

He, K. B., Yang, F. M., Ma, Y. L., Zhang, Q., Yao, X. H., Chan, C. K., Cadle, S., Chan, T., and Mulawa, P.: The characteristics of PM2.5 in Beijing, China, Atmos. Environ., 35, 4959–4970, doi:10.1016/S1352-2310(01)00301-6, 2001.

Horton, D. E. and Diffenbaugh, N. S.: Response of air stagnation frequency to anthropogenically enhanced radiative forcing, Environ. Res. Lett., 7, 044034, doi:10.1088/1748-9326/7/4/044034, 2012.

Jacob, D. J., and Winner, D. A.: Effect of climate change on air quality, Atmos. Environ., 43, 51–63, doi:10.1016/j.atmosenv.2008.09.051, 2009.

Liao, H., Chen, W. T., and Seinfeld, J. H.: Role of climate change in global predictions of future tropospheric ozone and aerosols, J. Geophys. Res.-Atmos., 111(D12), doi:10.1029/2005JD006852, 2006.

Yang, Y. Q., Wang, J. Z., Gong, S. L., Zhang, X. Y., Wang, H., Wang, Y. Q., Wang, J., Li, D., and Guo, J. P.: PLAM - a meteorological pollution index for air quality and its applications in fog-haze forecasts in North China, Atmos. Chem. Phys., 16, 1353-1364, doi:10.5194/acp-16-1353-2016, 2016.

*2. Page 3, line 13-17: For each of the other radiosonde stations: it is better to clarify how many station, for example 8 other radiosonde stations? According to Guo et al., ACP 2016, there are 120 radiosonde stations across China, and actually only part of the whole radiosonde network was used in this study.*
**Response:**

We obtained datasets of all the radiosonde stations across China (95 stations) and two stations (Blagovescensk and Vladivostok) outside but near the border of the country. Among them, 66 stations have corresponding surface datasets from CMA (See Appendix A). For each of the other 31 radiosonde stations without corresponding surface data, we considered the average of surface stations within 150 km as a substitute. In this way, we got additional 15 stations (See Appendix B). Air stagnations of these 81 stations are analyzed in this study. We have rewritten this part

of sentences in the revised the manuscript to clarify.

There are 120 upper-air sounding stations in China, except for Hong Kong and Taiwan district (Li, 2006; Guo et al., 2016). The stations used in our study are those participate in global data exchange, which are available from the free-access Wyoming University soundings database (http://weather.uwyo.edu/upperair/sounding.html).

Reference:

Guo, J. P., Miao, Y. C., Zhang, Y., Liu, H., Li, Z. Q., Zhang, W. C., He, J., Lou, M. Y., Yan, Y., Bian, L. G., and Zhai, P.: The climatology of planetary boundary layer height in China derived from radiosonde and reanalysis data, Atmos. Chem. Phys., 16, 13309-13319, doi:10.5194/acp-16-13309-2016, 2016.

Li F. New Development with Upper Air Sounding in China. WMO TECO, 2, 2006.

*3. Figure 1: The site outside of China like Blagovescensk, Vladivostok, and KingsPark should be plotted and marked.*

**Response:**

Blagovescensk and Vladivostok have been marked on Fig. 1 in the revised manuscript. Kings Park is a site in Hong Kong, and has been treated as a normal site in China.

*4. Page 4, line 25: What kind of spatial interpolation methods have the authors applied? More details should be given, like the size of grid point for the maps.*

**Response:**

Results of stagnation days and cases were interpolated with cubic splines to 2°×2° grid. This detail has been added in the revised manuscript.

*5. Section 3.2:" During summer a weaker pressure gradient lingers over China, which results in weaker atmospheric circulations and more frequent stagnations." I think at least one figure is needed to corroborate this argument, e.g., the pressure gradient difference between summer and annual mean.*

**Response:**

Pressure patterns vary in strength and location seasonally. This is a fact known in climate studies in China (e.g. Ding et al., 2013). Figure R1 shows that in upper layer, the gradient of geopotential height is much weaker in summer than that in winter, so the pressure gradient varies in the same way. At surface (sea level), in winter, a very strong high-pressure center (i.e. the Siberian high) is generated over the frozen landscape of northern Asia. In summer, high surface temperatures over the land generate lows that replace wintertime highs. In general, pressure gradient at surface is also weaker in summer than in winter over most of China.

We are sorry that we had not properly cited reference about this fact. Now, we add the citation and the fact description in the revised manuscript.

[Figure]

Figure R1. Atmospheric circulation in East Asia. Upper: geopotential height (unit: 10gpm) of 500 hPa in January (left) and July (right). Bottom: sea level pressure (unit: hPa) in January (left) and July (right). This figure is copied from Ding et al. (2013), their Fig. 1.1.

Reference:

Ding, Y. H., Wang, S. W., Zheng, J. Y., Wang, H. J., and Yang, X. Q.: Climate of China, Science Press, Beijing, 557pp, 2013 (in Chinese).

---

## Author Comment (AC2) · 10 Apr 2017

**Response to Referee #2**

Manuscript: Air Stagnations for China (1985–2014): Climatological

Mean Features and Trends (acp-2016-1072)

We are very grateful to the referees for their insightful comments. We carefully consider all the comments and suggestions, and carry out additional analysis on relevant problems. We hope that we have taken most of the referees' concerns into the revised manuscript. All the comments of Referee #2 are copied below in italics, and followed by our responses. Additional material containing two figures (Figs. R2 and R3 in this Response) is supplied in the Supplement.

**Referee #2**

*General comments:*
*This study explores the long-term surface meteorology and sounding measurements in China to study the climatology and inter-decadal trends of air stagnation in China since the 1980s. Considering the current severe haze pollution in China, this topic is interesting to the community and the compiled data sets are valuable for other pollution-related researches. Therefore, the materials in the manuscript are definitely publishable. However, my major concern is that, in contrast to the abundance of data involved in this study, the analyses of the data and the exploration of physics behind the observed relationships are quite inadequate. The authors should carefully address this issue before I can recommend its publication on ACP. Some detailed comments are provided below.*
**Response:**
    We thank the referee very much for the positive evaluation to this paper. We have added more introduction and discussion to improve the knowledge behind the phenomena in the revised manuscript. We hope that we have explored more physics behind air stagnation index and its trend in this revision.

*Specific comments:*
*1) Section 3.2, the seasonality of air stagnation is interesting, but the authors' explanation and analysis are insufficient. For example, authors simply attributed the peak of stagnation in the summer to the "weak pressure gradient", but failed to explain what cause the weak pressure gradient in the summertime Eastern China. How is it linked to the sub-tropical high or the monsoon circulation? Answers can be found by looking at reanalysis of meteorological fields.*
**Response:**
    This question is similar to Question 5 of Referee #1. Our explanation is based on a fact in climate studies of China (Ding et al. 2013), but we are sorry that we had not cited proper reference about it in the original manuscript.

A much weaker pressure gradient in summer is the seasonal feature in mid-latitudes (Frederick et al., 2012). This feature is very evident in upper layer atmosphere in China (Fig. R1, Ding et al. 2013). However at the sea-level surface, the case in eastern Asia and China are complicated by the sub-tropical high in the east and the continental low and in the west respectively. As a result, Asia summer monsoon prevails in eastern China. Though for this, the sea level pressure gradient in summer is still much weaker than that in winter (Fig. R1, Ding et al. 2013).

This explanation has been added in the revised manuscript.

[Figure]

Figure R1. Atmospheric circulation in East Asia. Upper: geopotential height (unit: 10gpm) of 500 hPa in January (left) and July (right). Bottom: sea level pressure (unit: hPa) in January (left) and July (right). This figure is copied from Ding et al. (2013), their Fig. 1.1.

Reference:

Ding, Y. H., Wang, S. W., Zheng, J. Y., Wang, H. J., and Yang, X. Q.: Climate of China, Science Press, Beijing, 557pp, 2013 (in Chinese).

Frederick, K. L., Edward, J. T., and Dennis, G. T.: The atmosphere: an introduction to meteorology, Prentice Hall, 528pp, 2012.

*2) Similarly, it is interesting to see the larger influence of upper-level wind on air stagnation than that of the surface wind (this point should be highlighted in the abstract), but the physical relationship between upper-level wind and air stagnation is vaguely described in the manuscript. It should be thoroughly discussed in the introduction part. Is the upper-level wind here associated with moving speed of high pressure systems?*

**Response:**

The "upper-air winds" in air stagnation index refer to winds at about 5 kilometers above the ground. From a meteorological perspective, this level is

important because of its connection to near-surface synoptic systems. It is found that the movement of surface cyclones tends to travel in the direction of the upper flow at roughly a quarter to half of the speed (Frederick et al., 2012). These kinds of near-surface synoptic systems are essential to air pollution (Jacob and Winner, 2009; Cai et al., 2017). This simply means, the weather system has large impacts on air stagnation.

This part of discussion has been added in the revised manuscript.

Reference:

Cai, W., Li, K., Liao, H., Wang, H., and Wu, L.: Weather conditions conducive to Beijing severe haze more frequent under climate change, Nat. Clim. Change, 7, 257–262, doi: 10.1038/nclimate3249, 2017.

Frederick, K. L., Edward, J. T., and Dennis, G. T.: The atmosphere: an introduction to meteorology, Prentice Hall, 528pp, 2012.

Jacob, D. J., and Winner, D. A.: Effect of climate change on air quality, Atmos. Environ., 43, 51–63, doi:10.1016/j.atmosenv.2008.09.051, 2009.

*3) Inter-decadal trends of air stagnation in China are derived in this study, but what drive these trends are not discussed in the paper. Greenhouse gas, anthropogenic aerosol, or some other forcing agents? I understand that this complex question may be beyond the main scope of the paper, but some discussions along this line could largely sharpen the climatic implication of this study. I suggest a more thorough literature review about recent climate changes in China, in terms of hydrological cycle and precipitation extremes (Wang Y. et al., 2016, JGR), monsoon circulations (Li Z., et al., 2016, Rev. Geos.), etc, and the authors can link the trends to the possible factors at play.*

**Response:**

We thank the referee for this suggestion. As mentioned by the referee, the climate system is complex, so is the climate change. Although the greenhouse gas and anthropogenic aerosol may provide a general background to global climate change, there are still large uncertainties in regional climate responses or trends. Our work shows the fact of trends only based on the air stagnation metric.

Three components of the air stagnation metric are: upper-air winds, near-surface winds and daily precipitation. Our results have shown that the increasing trend of air stagnation is mainly caused by the decreasing trend of upper-air winds. The decreasing trend of near-surface winds exerts a minor influence, while the increasing number of dry days has the least influence.

Studies have shown that wind speeds of mid- and upper-troposphere decrease in the past 30 years (Zhang et al., 2009). The trend can be attributed to global warming, which results in smaller contrasts of the sea level pressure, and near-surface temperature between the Asian continent and the Pacific Ocean. A consistent weakening and poleward expansion of the Hadley circulation in the climate change background also plays a role (Frierson et al., 2007; Hu et al., 2011; Jiang et al., 2010; Lau et al., 2006; Lu et al., 2007; Seidel et al., 2008).

Over China, near-surface wind decline is attributed to (1) slowdown in atmospheric general circulation and weakening in synoptic weather activity, and thus reduced downward transport of the horizontal momentum by the faster wind aloft (Guo et al., 2011; Jiang et al., 2010; Vautard et al., 2010; Xu et al., 2006). (2) increasing surface roughness in the near field of each station and/or in boundary layer structure (Vautard et al., 2010); (3) light absorbing aerosols, which stabilize the atmosphere by cooling the surface and warming the upper. As a result, the vertical flux of horizontal momentum is reduced (Erlick et al., 2003; Li et al., 2016; Pandithurai et al., 2008; Peng et al., 2016; Wang et al., 2013; Yang et al., 2013).

The decreasing number of rainy days (Li et al., 2016; Gong et al., 2004; Liu et al., 2002) also play a positive role in the increasing of air stagnation over China. According to Gong et al. (2004), such trend is mainly caused by the significant reduction of days with light rain. And accumulation of greenhouse gases and dramatic increases of anthropogenic aerosols may be responsible for that (Liu et al., 2015; Wang et al., 2016).

We have added this literature review in the revised manuscript.

References:

Erlick, C., and Ramaswamy, V.: Sensitivity of the atmospheric lapse rate to solar cloud absorption in a radiative-convective model, J. Geophys. Res.-Atmos., 108, doi:10.1029/2002JD002966, 2003.

Frierson, D. M. W., Lu, J., and Chen, G.: Width of the Hadley cell in simple and comprehensive general circulation models, Geophys. Res. Lett., 34, doi:10.1029/2007GL031115, 2007.

Gong, D. Y., Shi, P. J., and Wang, J. A.: Daily precipitation changes in the semi-arid region over northern China, J. Arid Environ., 59, 771-784, doi:10.1016/j.jaridenv.2004.02.006, 2004.

Guo, H., Xu, M., Hu, Q.: Changes in near-surface wind speed in China: 1969-2005, Int. J. Climatol., 31, 349-358, doi:10.1002/joc.2091, 2011.

Hu, Y., Zhou, C., and Liu, J.: Observational evidence for poleward expansion of the Hadley circulation, Adv. Atmos. Sci., 28, 33-44, doi:10.1007/s00376-010-0032-1, 2011.

Jiang, Y., Luo, Y., Zhao, Z., Tao, S.: Changes in wind speed over China during 1956–2004, Theor. Appl. Climatol., 99, 421-430, doi:10.1007/s00704-009-0152-7, 2010.

Lau, N. C., Leetmaa, A., and Nath, M. J.: Attribution of atmospheric variations in the 1997–2003 period to SST anomalies in the Pacific and Indian Ocean basins, J. Clim., 19, 3607-3628, doi:10.1175/JCLI3813.1, 2006.

Li, Z. Q., Lau, W. K. M., Ramanathan, V., Wu, G., Ding, Y., Manoj, M. G., Liu, J., Qian, Y., Li, J., Zhou, T., Fan, J., Rosenfeld, D., Ming, Y., Wang, Y., Huang, J., Wang, B., Xu, X., Lee, S. S., Cribb, M., Zhang, F., Yang, X., Zhao, C., Takemura, T., Wang, K., Xia, X., Yin, Y., Zhang, H., Guo, J., Zhai, P. M., Sugimoto, N., Babu, S. S., and Brasseur, G. P.: Aerosol and monsoon climate interactions over Asia, Rev. Geophys., 54, 866-929, doi:10.1002/2015rg000500, 2016.

Liu, R., Liu, S. C., Cicerone, R. J., Shiu, C. J., Li, J., Wang, J. L., and Zhang, Y. H.: Trends of Extreme Precipitation in Eastern China and Their Possible Causes, Adv. Atmos. Sci.,

32, 1027-1037, doi:10.1007/s00376-015-5002-1, 2015.

Liu, S. C., Wang, C. H., Shiu, C. J., Chang, H. W., Hsiao, C. K., and Liaw, S. H.: Reduction in sunshine duration over Taiwan: Causes and implications, Terr. Atmos. Ocean. Sci., 13, 523-545, 2002.

Lu, J., Vecchi, G. A., and Reichler, T.: Expansion of the Hadley cell under global warming, Geophys. Res. Lett., 34, 5, doi:10.1029/2006gl028443, 2007.

Pandithurai, G., Seethala, C., Murthy, B. S., and Devara, P. C. S.: Investigation of atmospheric boundary layer characteristics for different aerosol absorptions: Case studies using CAPS model, Atmos. Environ., 42, 4755-4768, doi:10.1016/j.atmosenv.2008.01.038, 2008.

Peng, J., Hu, M., Guo, S., Du, Z., Zheng, J., Shang, D., Zamora, M. L., Zeng, L., Shao, M., Wu, Y.-S., Zheng, J., Wang, Y., Glen, C. R., Collins, D. R., Molina, M. J., and Zhang, R.: Markedly enhanced absorption and direct radiative forcing of black carbon under polluted urban environments, Proc. Natl. Acad. Sci. U. S. A., 113, 4266-4271, doi:10.1073/pnas.1602310113, 2016.

Seidel, D. J., Fu, Q., Randel, W. J., and Reichler, T. J.: Widening of the tropical belt in a changing climate, Nat. Geosci., 1, 21-24, doi:10.1038/ngeo.2007.38, 2008.

Vautard, R., Cattiaux, J., Yiou, P., Thepaut, J.-N., and Ciais, P.: Northern Hemisphere atmospheric stilling partly attributed to an increase in surface roughness, Nat. Geosci., 3, 756-761, doi:10.1038/ngeo979, 2010.

Wang, Y., Khalizov, A., Levy, M., and Zhang, R. Y.: New Directions: Light absorbing aerosols and their atmospheric impacts, Atmos. Environ., 81, 713-715, doi:10.1016/j.atmosenv.2013.09.034, 2013.

Wang, Y., Ma, P.-L., Jiang, J. H., Su, H., and Rasch, P. J.: Toward reconciling the influence of atmospheric aerosols and greenhouse gases on light precipitation changes in Eastern China, J. Geophys. Res.-Atmos., 121, 5878-5887, doi:10.1002/2016jd024845, 2016.

Xu, M., Chang, C. P., Fu, C. B., Qi, Y., Robock, A., Robinson, D., and Zhang, H. M.: Steady decline of east Asian monsoon winds, 1969-2000: Evidence from direct ground measurements of wind speed, J. Geophys. Res.-Atmos., 111, doi:10.1029/2006jd007337, 2006.

Yang, X., Yao, Z., Li, Z., and Fan, T.: Heavy air pollution suppresses summer thunderstorms in central Chin J. Atmos. Sol.-Terr. Phy., 95-96, 28-40, doi:10.1016/j.jastp.2012.12.023, 2013.

Zhang, A. Y., Ren, G. Y., Guo, J., and Wang, Y.: Change trend analyses on upper-air wind speed over China in past 30 years, Plateau Meteor, 28, 680-687, 2009 (in Chinese).

*4) Would the increasing trend of air stagnation be partly caused by the accumulation of the absorbing aerosols in boundary layer and a consequent stabilization effect (Wang Y., et al., 2013, AE; Peng J. et al., 2016, PNAS)?*
**Response:**
Our response for 3) has covered this question.

*5) The correlation analyses in Fig. 8 and 10 use annual mean values. Why not monthly mean? That would give you more dots and larger statistical significance.*

**Response:**

As suggested, we also use seasonal mean values to conduct correlation analyses (Figs. R2 and R3). According to Fig. 8, the spatial distribution of dry days barely correlates with that of air stagnations. So we did not further investigate the seasonal difference of correlation between them. Figures R2 and R3 are presented in the supplement as Fig. S1 and Fig. S2.

[Figure]

Figure R2. Seasonal dependence of spatial distribution of stagnation days on upper-air wind speed and surface wind speed. Linear regression coefficients between seasonal-mean stagnation days at 81 stations and each corresponding component are shown. Green: spring (MAM); red: summer (JJA); orange: autumn (SON); black: winter (DJF).

[Figure]

Figure R3. Same as Fig. R2, but for the seasonal trends of stagnant days. Linear regression coefficients between national-averaged stagnant days in different seasons over 30-year period and corresponding components (upper-air wind speed, surface wind speed and dry days) are shown. Green: spring (MAM); red: summer (JJA); orange: autumn (SON); black: winter (DJF).

*6) What is the correlation between the air stagnation and visibility measurements in China since the 1980s? I assume they are highly correlated, but it is still interesting to see their relationships in time series and spatial distribution in China.*

**Response:**

This question is similar to Question 1 of Referee #1. We carry out additional analysis on this topic.

30-year (1985–2014) visibility data of 360 stations across China were obtained from NCDC (U.S. National Climatic Data Center). Figure R4 displays the spatial distribution of the stations and annual mean visibility. It shows that generally the east and south regions of China exhibit poor visibility, while the west and north regions exhibit a good one. Regions of Sichuan basin, the west of Xinjiang and North China Plain are the centers exhibit low visibility. This feature corresponds well to the frequent air stagnation occurrences in Fig. 2. Yangtze River Delta also exhibits poor visibility, but shows relatively less stagnant days (Fig. 2). The correlation between the air stagnation days and a time series of visibility over the whole country is −0.69 during 1985–2014 (Fig. R5). It means that the air stagnation does correlate negatively to visibility, in general.

Air stagnation is a simple and meaningful metric to air pollution. Jacob and Winner (2009) have summarized results from different studies about the influence of meteorological factors on ozone and particulate matter concentrations, and concluded that air stagnation consistently demonstrates a strong positive correlation. In this paper, we are trying to present the climatological features of air pollution potential over China by analyzing stagnation index, rather than directly applying to air pollution diagnosis or forecasting. Therefore, at this stage, we do not emphasize the correlation between the metric and visibility, and this part of results are not included in the revised manuscript.

[Figure]

Figure R4. Spatial distribution of (a) visibility observation stations and (b) annual mean visibility throughout China (1985–2014).

[Figure]

Figure R5. Time series of national averaged stagnant days and visibility during 1985–2014. Solid line: stagnation days; dashed line: visibility. Linear regression correlation between them are shown.

Reference:

Jacob, D. J., and Winner, D. A.: Effect of climate change on air quality, Atmos. Environ., 43, 51–63, doi:10.1016/j.atmosenv.2008.09.051, 2009.

---

## Author Response (AR3)

**Response to Editor**

**Manuscript: Air Stagnations for China (1985–2014): Climatological**

**Mean Features and Trends (acp-2016-1072)**

We are very grateful for the editor's comments and the manuscript has been revised accordingly. We have also carefully inspected the language usage in the current version and hope that its readability has been improved. All the editor's comments are copied below in italics, and followed by our responses. The new changes in the manuscript are marked in green color.

*Comments to the Author:*
*I still notice several issues with the language usage in the current version and ask for additional effort to improve its readability. Below are a few examples.*

*Abstract, 1st line, the phrase is awkward "Air stagnation is an important meteorological measurement". Change "measurement" to "measure" or "reflection".*
*The next sentence "We conducted a comprehensive investigation of air stagnation in China, based on sounding and surface observations of 81 stations, from January 1985 to December 2014." to "We conducted a comprehensive investigation of air stagnation in China from January 1985 to December 2014, based on sounding and surface observations of 81 stations".*
*Change "The stagnation criteria were revised to be topographically dependent for the great physical diversity in this country" to "The stagnation criteria were revised to account for the large topographical diversity in this country".*
**Response:** Accepted.

*The phrase "Dependence degrees" is confusing.*
**Response:**
     "Dependence degrees of air stagnations on three components (upper- and lower-air winds, precipitation-free days) are examined" in the Abstract has been changed to "Changes in air stagnation occurrences are dependent on three components (upper- and lower-air winds, precipitation-free days)".
     "The dependence degrees of national averaged stagnation trends" in page 8 line 25 has been changed to "The dependence of national averaged stagnation trends".

*Page 3, line 18, please avoid using the progressive tense "Our focus in the current study is investigating".*
**Response:** The sentence has been changed to "Our primary purpose in the current study is to investigate ..."

*Also, I believe that it is important to include some discussion on the possible effects of the aerosol indirect effect on air stagnation in China. It has been clearly shown that long-term cloud properties and precipitation have been significantly modified by highly elevated aerosols in China, as evident by suppressed light rainfall but intensified heavy rainfall (Wang et al., Long-term impacts of aerosols on precipitation and lightning over the Pearl River Delta megacity area in China, Atmos. Chem. Phys. 11, 12421, 2011).*

[revised manuscript text omitted]